# Restoring Negative Information in Few-Shot Object Detection

**Yukuan Yang**\*
Tsinghua University
`yyk17@mails.tsinghua.edu.cn`

**Fangyun Wei**
Microsoft Research Asia
`fawe@microsoft.com`

**Miaojing Shi**
King's College London
`miaojing.shi@kcl.ac.uk`

**Guoqi Li**
Tsinghua University
`liguoqi@mail.tsinghua.edu.cn`

## Abstract

Few-shot learning has recently emerged as a new challenge in the deep learning field: unlike conventional methods that train the deep neural networks (DNNs) with a large number of labeled data, it asks for the generalization of DNNs on new classes with few annotated samples. Recent advances in few-shot learning mainly focus on image classification while in this paper we focus on object detection. The initial explorations in few-shot object detection tend to simulate a classification scenario by using the positive proposals in images with respect to certain object class while discarding the negative proposals of that class. Negatives, especially hard negatives, however, are essential to the embedding space learning in few-shot object detection. In this paper, we restore the negative information in few-shot object detection by introducing a new negative- and positive-representative based metric learning framework and a new inference scheme with negative and positive representatives. We build our work on a recent few-shot pipeline RepMet [1] with several new modules to encode negative information for both training and testing. Extensive experiments on ImageNet-LOC and PASCAL VOC show our method substantially improves the state-of-the-art few-shot object detection solutions. Our code is available at `https://github.com/yang-yk/NP-RepMet`.

## 1  Introduction

In the past decade, there has been a transformative revolution in computer vision cultivated by the adoption of deep learning [2]. Driven by the increasing availability of large annotated datasets and efficient training techniques, deep learning-based solutions have been progressively employed from image classification to action recognition. The majority of deep learning methods are designed to solve fully-supervised problems where large amount of data come with carefully assigned labels. In contrast, humans, even children can easily recognize a multitude of objects in images when told only once or few times, despite the fact that the image of objects may vary in different viewpoints, sizes and scales. This ability, however, is still a challenge for machine perception.

To enable machine perception with only few training samples, some studies start shifting towards the so-called few-shot learning problem: after learning on a set of base (seen) classes with abundant examples, new tasks are given with only few support images of novel (unseen) classes. Recent advances in few-shot learning mainly focus on image classification and recognition tasks [3–9].

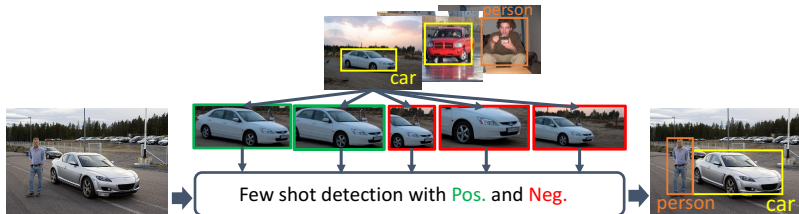

Figure 1: Restoring negative information in few-shot object detection.

Nonetheless, few-shot learning can also be applied to more complex tasks, e.g. object detection [10, 1, 11–13], assuming bounding box annotations are available in few support images for new classes. Investigation further this line is very limited. Initial explorations resemble solutions in the few-shot classification [3, 4], where prototype representations are learned [1] and weighted [10, 11] from the few labeled samples per class, and used to match the query sample of a specific class.

For the convenience of adapting few-shot classification methods, the common practice in few-shot object detection [10, 1, 12, 11, 13] directly extracts positive proposals (green boxes in Figure. 1) of large Intersection over Union (IoU) with ground truth (yellow) from support images, while discards negative proposals (red) containing partial objects, ambiguous surrounds, or complex backgrounds in images. As a result, these negative proposals often end up as false positives in the final detection (see Figure. 3). In the meantime, negative proposals in fully-supervised object detection [14–16] are carefully evaluated via their IoU with ground truth; hard negatives (e.g. $\tau$<IoU<$t$ ) are particularly selected to train against with and improve the robustness of the classifier.

Building upon the above observation, the purpose of this study is to restore the negative information properly in few-shot object detection. Our essential idea is to make use of both positive and negative proposals in training images (Figure. 1): an embedding space can be learnt upon them where distances correspond to a measure of object similarity to both positive and negative representatives. Once this space is learnt, few-shot object detection can be easily implemented using any standard techniques with our proposed embedding method as feature vectors. Without loss of generality, we build our work on top of an established pipeline, RepMet [1], where multiple positive representatives are learnt for each base class at training, and replaced by embedding vectors from positive proposals of support images for new classes at testing. In light of the importance of negative information in images, we propose to split the class representation in RepMet into two modules to learn negative and positive representatives separately; the embedding vector of a given proposal is also replaced with a new negative and positive embedding (NP-embedding). The optimization of the embedding space differs between negative and positive proposals: if a proposal is positive to a certain class, we want to push it close to those positive representatives of that class and away from those negative representatives of that class; if it is negative to a certain class, the optimization is the opposite. We introduce triplet losses based on the NP-embedding for this purpose. The class label prediction branch in RepMet is also adapted with the proposed NP-embedding.

At the inference stage with new classes, the learnt representatives are replaced with embedding vectors from both positive and negative proposals harvested in supported images. The number of negative proposals is much more than that of positive proposals. To select hard and diverse negatives, we first choose them with an IoU criterion ($\tau$<IoU<$t$) w.r.t ground truth; a clustering-based selection strategy is further introduced to guarantee the diversity of negative proposals.

To our knowledge, we are the first to present the idea of restoring negative information in few-shot object detection. The contribution of our work is three-fold:

- We introduce a new negative- and positive-representative based metric learning framework (NP-RepMet) with negative information incorporated in different modules for better feature steering in the embedding space.
- We propose a new inference scheme using both negative and positive representatives; for harvesting the hard and diverse negatives from the support set, a clustering-based hard negative selection strategy is also introduced.
- Extensive experiments on standard benchmarks ImageNet-LOC [1] and PASCAL VOC 2007 [10] demonstrate that our method substantially improves the SOTA (i.e. up to +11% on ImageNet-LOC and +19% on PASCAL VOC).

## 2 Related Works

**Few-shot learning.** Few shot learning is not a new problem: its target is to recognize previously unseen classes with very few labeled samples [17–22]. The recent resurgence in interest of few-shot learning is through the so-called meta-learning [23–25, 20, 4], where *meta-learning* and *meta-testing* are performed in a similar manner; representative works in image classification include matching network [4] and prototypical network [3]. Apart from meta-learning, some other approaches make use of sample synthesis and augmentation in few-shot learning [26, 5, 27, 28].

**Few-shot object detection.** In contrast to classification, few-shot object detection is not largely explored. Karlinsky et al. [1] introduce an end-to-end representative-based metric learning approach (RepMet) for few-shot detection; Kang et al. [10] present a new model using a meta feature learner and a re-weighting module to fast adjust contributions of the basic features to the detection of new classes. Fan et al. [13] extend the matching network by learning on image pairs based on the Faster R-CNN framework, which is equipped with multi-scale and shaped attentions. Some other works modelling the meta-knowledge based on Faster R-CNN can be found in [12, 11]. These approaches fall within the meta-learning regime. Whilst there exist many other works trying to solve the problem from the domain transfer/adaption perspective [29, 30]. For instance, Chen et al. [29] propose a low-shot transfer detector (LSTD) to leverage rich source-domain knowledge to construct a target-domain detector with few training examples. Transfer learning in [29, 30] requires training on both source (base) and target (new) classes. Meta-learning instead can be more efficient in the sense its predication on new classes can be directly achieved via network inference. In this paper, we focus on the meta-learning.

**Comparison to RepMet.** Our work is built on RepMet [1] but substantially improves it with the restoration of negative information at both training and inference stages. It should be noted that negative information has been used in RepMet similar to the usage of negatives in few-shot image classification: class representatives from different classes are considered as negatives to each other; online hard example mining (OHEM) [31] is also adopted. These negatives are collected across images, we instead bootstrap the classifier with negatives both within and across images. Mining negatives within the same image of positives is rather standard for fully supervised object detection [14, 15, 32, 33], as it provides a better feature steering in the embedding space. We believe this essential idea should also apply to the few-shot object detection.

## 3 Method

### 3.1 Overview

**RepMet.** Some core modules of RepMet [1] are illustrated in Figure. 2 with light green background. It learns positive class representatives $\{R^p_{ij}|1 \leq i \leq N, 1 \leq j \leq K\}$ as weights of an FC layer of size $N \cdot K \cdot e$, where $i$ and $j$ denote the $i$-th class and $j$-th representative. $N$ is the number of classes, $K$ is the total number of representatives per class, and $e$ denotes the dimensionality of each representative. In the training stage, a given foreground (positive, e.g. IoU>0.7 in Figure. 2) proposal is embedded through the DML embedding module as a vector $E^p$. The network computes distance from $E^p$ to representatives $R^p_{ij}$. The distances are optimized with 1) a cross entropy loss to predict the correct class label; 2) an embedding loss to enforce a margin between the distance of $E^p$ to the closest representative of the correct class and the closest representative of a wrong class.

**Negative information restoration.** We build our work on RepMet with negative and positive information, and name it NP-RepMet in Figure. 2. At training stage, apart from positive representatives ($R^p_{ij}$), negative representative ($R^n_{ij}$) are also learnt with another FC layer of size $N \cdot K \cdot e$. Given an object proposal $P$ from RPN, we modify the original DML embedding module from [1] to branch off two vectors ($E^n$ and $E^p$) to learn $R^n_{ij}$ and $R^p_{ij}$ separately. $P$ is categorized as either positive or negative proposal according to its IoU with the ground truth. If $P$ is positive to class $i$, only its positive embedding $E^p$ is used to learn $R^p_{ij}$; if $P$ is negative to class $i$, only $E^n$ is used vice versa. Different embedding loss functions are proposed for the two scenarios. Both $E^n$ and $E^p$ are used to compute the class posterior probability of $P$ in a form of a cross entropy loss to the ground truth label. When testing with new classes, the learnt $R^n_{ij}$ ($R^p_{ij}$) are replaced with the negative (positive) embedding vectors $E^n$ ($E^p$) of negative (positive) proposals from support images.

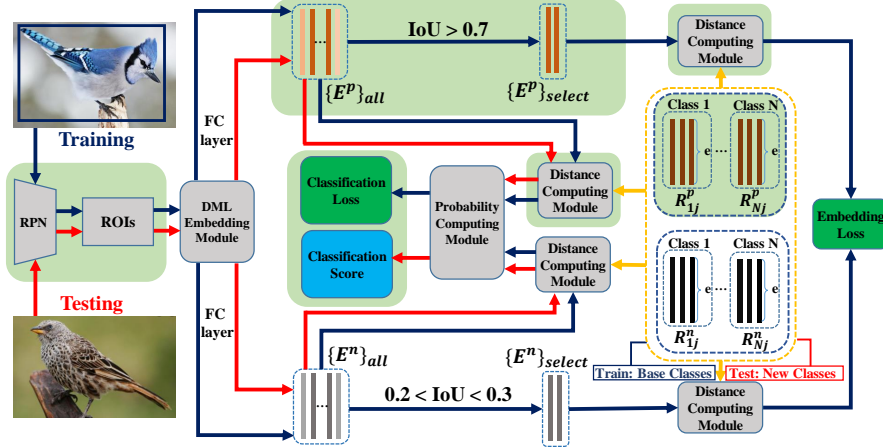

Figure 2: Overview of NP-RepMet. The blue and red line signifies the training and testing flow, respectively. Yellow box denotes sets of negative and positive representatives ($R^n$ and $R^p$) which are learnt on base classes with negative and positive embedding ($E^n$ and $E^p$) of proposals from training images. They are replaced by embedding vectors of support images for new classes at testing.

## 3.2 Negative- and Positive-Representative Based Metric Learning

The essential idea of this work is to restore negative information in the few-shot learning pipeline and learn the embedding space from both negative and positive information. Applying this idea to RepMet offers several new modules (Figure. 2):

**Negative and positive representatives.** Apart from positive representatives ($R_{ij}^p$) in RepMet, another FC layer for negative representatives ($R_{ij}^n$) are delivered. Two sets of representatives will therefore be learnt for each class. Both $R^n$ and $R^p$ are randomly initialized. They are learnt with different information.

**Negative and positive proposals.** Given proposals produced by RPN, we separate positive and negative proposals ($P$) according to their IoU w.r.t the ground truth $G$. Concretely, we take those of IoU($P$, $G$) > 0.7 as positives and those of 0.2 < IoU($P$, $G$) < 0.3 as negatives.

**Negative and positive embedding (NP-embedding).** Given an object proposal $P$ from a training image, instead of embedding it into a single vector, we embed it into two vectors ($E^n$ and $E^p$) to learn $R_{ij}^n$ and $R_{ij}^p$ separately. This is achieved by branching off another convolutional layer after the second last layer of the DML embedding module in RepMet. This separated embedding allows faster and more optimal convergence of the learning on $R_{ij}^n$ and $R_{ij}^p$.

With the availability of above modules, we define new triplet losses to learn $R_{ij}^n$ and $R_{ij}^p$.

**Triplet losses based on NP-embedding.** We treat positive and negative proposals separately to learn the embedding space for $R_{ij}^p$ and $R_{ij}^n$. Given a positive proposal $P$ of class $i^*$, we have two distances for it: 1) the distance from its positive embedding vector $E^p$ to its closest positive representative $R_{i^*j}^p$ of the same class; 2) the distance from its positive embedding vector $E^p$ to its closest negative representative $R_{i^*j}^n$ of the same class. The former should be smaller than latter. We define a triplet loss accordingly:

$$L(E^p, P) = |\min_j d(E^p, R_{i^*j}^p) - \frac{1}{2}(\min_j d(E^p, R_{i^*j}^n) + \min_{j,i \neq i^*} d(E^p, R_{ij}^p)) + \alpha|_+, \quad (1)$$

where $d(\cdot, \cdot)$ denotes the Euclidean distance, and $|\cdot|_+$ is the ReLu function; $\min_{j,i \neq i^*} d(E^p, R_{ij}^p)$ is inherited from RepMet: $R_{ij}^p$ from a different class of $i^*$ is also taken as a useful negative if it has the closest distance to $E^p$ over the positive representatives of all the other classes (similar to the usage in a classification task). Following [1], we ensure an $\alpha$ margin in Equation (1). Positive proposals of other object classes (e.g. bicycle, aeroplane, etc.) are mostly easy negatives to the current class (e.g. car), as they have different appearances. In contrast, negative proposals from images of the current class (e.g. car) are harder as they could contain partial, occluded, or entire object of the class (Figure. 1). Adding these hard negatives into the model learning results in a more robust classifier.

Similarly, if $P$ is a negative proposal, terms of "positive" and "negative" in the above distances are swapped. Its loss function becomes,

$$L(E^n, P) = |\min_j d(E^n, R^n_{i^*j}) - \frac{1}{2}(\min_j d(E^n, R^p_{i^*j}) + \min_{j,i\neq i^*} d(E^n, R^n_{ij})) + \alpha|_+, \qquad (2)$$

$\alpha$ is set to 0.5 as the same to [1] for both Equation (1) and (2).

Note that for a given proposal $P$, either the positive $E^n$ or negative $E^p$ embedding is used for learning the representatives. In the next, we will present a new *probability computing* module where both $E^n$ and $E^p$ of $P$ are used for its label prediction.

**Probability computing based on NP-embedding.** The *probability computing* module is responsible for the label prediction of $P$ and is optimized with the cross entropy loss. In RepMet, this module computes the upper bound of the real class probability by taking the minimal Euclidean distance of $d(E^p, R^p_{ij})$, $\min_j d(E^p, R^p_{ij})$, over all the $K$ modes of $R^p_{ij}$ for class $i$. Considering the fact that ground truth will not be available at test time, the cross entropy loss in NP-RepMet should be optimized with both $E^n$ and $E^p$. Following the same logic in [1], we compute the minimum of $d(E^p, R^p_{ij})$ and $d(E^n, R^n_{ij})$ and define the class probability as:

$$p_i(E^p, E^n) \propto \exp\left(-\frac{\min_j d(E^p, R^p_{ij}) - \beta \min_j d(E^n, R^n_{ij}) + 2\beta}{2\sigma^2}\right) \qquad (3)$$

Distances are mapped to a probability $p_i(E^p, E^n)$ using a Gaussian function like in [1]. Parameter $0 < \beta < 1$ is introduced to give a higher credit for the positive distance in (3). Each distance is computed with normalized feature vectors which results in a value $\in [0, 2]$, $2\beta$ is thus added to make sure the distance subtraction to be non-negative. $\beta$ is empirically chosen as 0.3. If $P$ is a positive proposal for class $i$, $p_i(E^p, E^n)$ should be big; otherwise, it should be small.

The overall loss function is a combination of the class cross entropy loss and triplet losses.

### 3.3 Inference with Negative and Positive Representatives

Figure. 2 illustrates the inference work flow in red. At inference, new classes are given with a small support set of labeled data. We follow the same procedure with RepMet to extract positive proposals. As for negative proposals, there exists a substantial amount of them, we introduce a clustering-based selection strategy to find diverse and hard negatives.

**Clustering-based hard negative selection.** Similar to Sec. 3.2, for a given class and its support images, we keep those hard negatives whose IoU with ground truth is between 0.2 and 0.3 as potential candidates. Next, in order to select the most diverse ones from these candidates, we introduce a clustering-based method: given negative embedding vectors $E^n_1, ..., E^n_M$ for hard negative proposals $P_1, ..., P_M$, we compute an $M \times M$ affinity matrix $S$ with elements $s_{ij} = E^n_i \cdot E^n_j$ being the dot product (feature similarity) between $E^n_i$ and $E^n_j$, where $i, j = 1, ..., M$. Given $S$, we apply the spectral clustering [34] onto it to obtain $K$ clusters. Proposals within each cluster are similar while across clusters are diverse. The most representative proposal from each cluster should be the centroid one that has the minimal average distance to others within the cluster. We select the $K$ centroid proposals as our hard negatives. Notice that after the filtering through the IoU constraint, the number of negative proposals has been substantially reduced to e.g. a few dozens. Spectral clustering can be quickly solved on such a small scale.

Given the negative and positive proposals selected from the support images, we embed them into the network to obtain vectors to replace the learnt negative and positive representatives, respectively. When a query image comes, we embed each of its proposal with NP-embedding in NP-RepMet and follow Equation (3) to infer its class probability.

## 4   Experiments

### 4.1   Dataset

We first evaluate our method on the benchmark established in [1] for a fair comparison with RepMet. Second, we evaluate our method in the same setup with [10] in the standard detection benchmark

Table 1: Results on ImageNet-LOC. Left: comparison with RepMet and baseline-FT in 1, 5 and 10-shot detection. Right: ablation study of NP-embedding (top) and NP-inference (bottom) in 1-shot detection.

| Dataset | Method | 1-shot | 5-shot | 10-shot |
|---|---|---|---|---|
| ImageNet-LOC | baseline-FT | 35.0 | 51.0 | 59.7 |
| (214 unseen | RepMet | 56.9 | 68.8 | 71.5 |
| animal classes) | Ours | **68.5** | **75.0** | **76.3** |
| ImageNet-LOC | RepMet | 86.0 | 90.2 | 90.5 |
| (100 seen animal classes) | Ours | **93.7** | **94.0** | **95.3** |

| Embedding | Single | NP |
|---|---|---|
| mAP | 65.8 | **68.5** |

| Train/Inference | Pos | NP |
|---|---|---|
| RepMet | 56.9 | 59.4 |
| NP-RepMet | 57.4 | **68.5** |

PASCAL VOC [35]. For classes in the ImageNet-LOC benchmark, they are mostly animals and birds species. 100 classes are selected as base (seen) classes for training while 214 classes are considered as new (unseen) classes for testing. Following [1], we adopt its 5-way $K \in \{1, 5, 10\}$ shot few-shot detection setting. For benchmark PASCAL VOC 2007, 15 out of 20 VOC classes are selected for training, the rest 5 are for testing. We use same splits as in [10, 12, 11] and carry out $K \in \{1, 2, 3, 5, 10\}$ shot detection.

## 4.2 Implementation Details and Evaluation Protocol

**Training details.** For ImageNet-LOC, we follow [1] to select 200 images from each base class for balanced training. For PASCAL VOC 2007, we follow [10] to use VOC 07 and 12 train/val sets for training. We use ResNet-101 [36] as backbone with DCN [37], feature pyramid network (FPN) [38] is employed as RPN to generate object proposals with six object scales. Top-2000 ROIs from the RPN are selected by OHEM. Backbone weights are pre-trained on COCO following [1] for ImageNet-LOC and pre-trained on ImageNet following [10] for PASCAL VOC. Other modules, e.g. FPN, RPN, DML, NP-Representatives etc., are randomly initialized. Our network is trained with synchronized stochastic gradient descent (SGD) over 4 GPUs with mini-batch of 4 images (1 image per GPU). The total epoch number is 20 and the learning rate is initialized as 0.01 and then divided by 10 at epochs 4, 6 and 15. The weight decay and momentum parameters are set as $10^{-4}$ and 0.9, respectively.

**Testing details.** We test the proposed method on new classes without performing any fine-tuning on both the ImageNet-LOC and PASCAL VOC benchmarks. We forward the images in support set to obtain the corresponding positive and negative representatives in the network and then forward the images in query set for detection. Testing on ImageNet-LOC is organized in episode of multiple new classes [1] while for PASCAL VOC we use the published snapshot of query and support samples from [10] for testing. NMS with threshold 0.7 is used to eliminate duplicated proposals generated by RPN. The top-2000 proposals will be used for category and location prediction. Last, soft-NMS [39] with threshold 0.6 is applied on the output as post-processing to merge duplicated bounding boxes.

**Evaluation protocol.** We adopt the most commonly used mean average precision (mAP) to evaluate the performance of few-shot object detection. A correct detection should have more than 0.5 IoU with the ground truth. We report mAP on the test set of ImageNet-LOC [1] and VOC 2007 [10, 12, 11].

## 4.3 Results on ImageNet-LOC

**Comparison with RepMet and other baselines.** We follow the same setup with RepMet to report NP-RepMet with 1-shot, 5-shot and 10-shot in Table 1-Left. The results for RepMet are 56.9, 68.8 and 71.5, respectively. By restoring negative information into RepMet, NP-RepMet significantly improves the results to 68.5, 75.0, and 76.3. In particular with the 1-shot scenario where the support for each class is very limited, our method provides an efficient way to mine useful negative formation within the support image, and we improve RepMet up to 11.6%! The margin of improvement gets smaller with 10-shot as the support set becomes more diverse.

There are also several baselines worth of comparison to NP-RepMet: for instance, we can train a standard object detector on base classes using the same FPN-DCN backbone and then fine-tune its classifier head on novel classes. This is denoted as 'baseline-FT' in [1] and Table 1: the reported results are 35.0, 51.0 and 59.7 in 1, 5 and 10-shot, respectively. More baseline implementations can be found in [1], they perform much inferior to RepMet/NP-RepMet.

Table 2: Results on ImageNet-LOC 1-shot setting. Left: ablation negative proposal selection at inference. Right: parameter variations of $\beta$ (top) and IoU (bottom) for hard negatives.

| Strategy | mAP | $\beta$ | 0.0 | 0.1 | 0.2 | 0.3 | 0.4 | 0.5 |
|---|---|---|---|---|---|---|---|---|
| RD | 66.5 | mAP | 54.3 | 67.0 | **68.5** | **68.5** | 68.2 | 68.0 |
| Cluster-RD | 67.1 | IoU Interval | (0, 0.1) | | (0.1, 0.2) | | (0.2, 0.3) | (0.3, 0.4) |
| Cluster-Min | **68.5** | mAP | 68.0 | | 68.3 | | **68.5** | 58.6 |

RepMet also reports results on the seen (base) classes, where they create episodes for the 100 seen classes and test them following the same 1, 5, and 10-shot. Since the episodes they use for seen classes are not published, we create our own episodes by randomly selecting 200 episodes for the 100 classes and report the results for both RepMet and NP-RepMet in Table 1. It can be seen that NP-RepMet maintains a good detection performance on base classes with mAP being 93.7, 94.0, and 95.3 in 1, 5, and 10-shot. Our results are higher than those of RepMet (86.0, 90.2 and 90.5).

**Ablation study.** All our ablation experiments are conducted on ImageNet-LOC in 1-shot. One basic module of our NP-RepMet is to add *negative representatives* ($R^n$ in Figure. 2) into RepMet, other modules are built upon this one. Without this basis, NP-RepMet collapses to RepMet. Results between NP-RepMet and RepMet are shown in Table. 1 where NP-RepMet significantly improves the mAP up to 11.6%. Negative representative is the cornerstone of our NP-RepMet. On top of it, we further ablate the importance of other modules.

*NP-embedding.* An object proposal in NP-RepMet is embedded into two vectors ($E^n$ and $E^p$ in Figure. 2) to learn the negative and positive representatives ($R_{ij}^n$ and $R_{ij}^p$). We may use only one embedding vector $E$ to learn both $R_{ij}^n$ and $R_{ij}^p$; subbranches of $E_{all}^n$ and $E_{all}^p$ in Figure. 2 will be merged together as $E$ while the remaining part is still split according to the proposal IoU with ground truth. Results in Table 1-Right (top) show that using single embedding vector produces mAP 65.8 vs. 68.5 of using two embedding vectors (NP-embedding). The proposed NP-embedding allows faster and more optimal convergence of the negative and positive representative learning.

*NP-representatives for inference.* To justify the usage of negative and positive representatives at inference (NP-inference), we compare it with the experiment using only positive representatives for inference (Pos-inference). The model can be trained either in the NP-RepMet or RepMet mode and we show both results in Table 1-Right (bottom). In the former, the result of Pos-inference is 57.4, which is much lower than that (68.5) of NP-inference. In the latter, Pos-inference becomes a normal RepMet with mAP 56.9. While for NP-inference, negative and positive proposals from support images are selected following the proposed way in the paper, but their embedding function is the same as in RepMet; simply adding negative representatives at the inference stage, NP-inference gives +2.5% improvement for RepMet. Since the embedding function for negatives and positives is not separately learnt, the performance of this NP-inference is still much lower than that (68.5) of the normal NP-RepMet. This justifies the importance of learning negative and positive representatives with NP-embedding in the training stage.

*Negative proposal selection at inference.* The number of negative proposals in support images can be more than what we want. Instead of randomly selecting them, we introduced a clustering-based selection strategy to select the hard and diverse negatives. It applies spectral clustering to the affinity matrix of feature similarities among negative proposals, and then find the centroid proposal within each cluster that has the minimal distance to others. We denote it by Cluster-Min. To demonstrate the effectiveness of Cluster-Min, we compare it with random selection (RD) in Table 2 Left: Cluster-Min is 2% higher than RD. In addition, we also present another result by using spectral clustering yet randomly choosing a proposal within each cluster. This (Cluster-RD) gives an mAP of 67.1, 1.4% lower than Cluster-Min, which further justifies the usage of centroid proposal.

**Parameter variations.** Next, we study the parameter variations in NP-RepMet.

$\beta$ *in probability computing module.* The class posterior probability of a given object proposal is computed based on its distance to both positive and negative representatives (Eq.3). We vary the parameter $\beta$ in Equation (3) and report the results in Table 2-Right (top). It can be seen that the optimal performance (68.5) occurs when $\beta$ is 0.3. A smaller $\beta$ is preferable to give a bigger weight on the distance from the positive embedding vector to positive class representatives.

*IoU for negative proposals.* Referring to Sec. 3.2, negative representatives are specifically learnt from negative proposals whose IoU with ground truth bounding box is between 0.2 ($\tau$) and 0.3 ($t$). We also

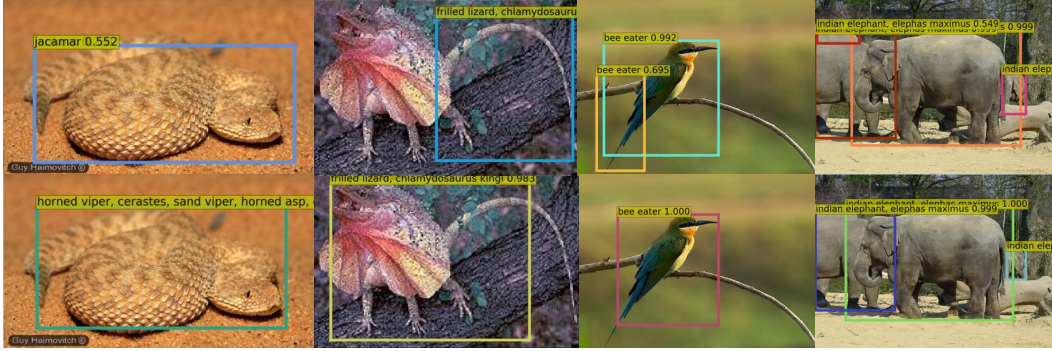

Figure 3: Detection results of RepMet (Top) and NP-RepMet (Bottom). The predicted label with confidence score is given for each proposal.

Table 3: Performance on PASCAL VOC 2007 novel classes.

| Method/Shot | novel set 1 | | | | | novel set 2 | | | | | novel set 3 | | | | |
|---|---|---|---|---|---|---|---|---|---|---|---|---|---|---|---|
| | 1 | 2 | 3 | 5 | 10 | 1 | 2 | 3 | 5 | 10 | 1 | 2 | 3 | 5 | 10 |
| YOLO-FR [10] | 14.8 | 15.5 | 26.7 | 33.9 | 47.2 | 15.7 | 15.3 | 22.7 | 30.1 | 39.2 | 19.2 | 21.7 | 25.7 | 40.6 | 41.3 |
| Meta-Det [12] | 18.9 | 20.6 | 30.2 | 36.8 | 49.6 | 21.8 | 23.1 | 27.8 | 31.7 | 43.0 | 20.6 | 23.9 | 29.4 | 43.9 | 44.1 |
| Meta R-CNN [11] | 19.9 | 25.5 | 35.0 | 45.7 | **51.5** | 10.4 | 19.4 | 29.6 | 34.8 | 45.4 | 14.3 | 18.2 | 27.5 | 41.2 | **48.1** |
| RepMet [1] | 26.1 | 32.9 | 34.4 | 38.6 | 41.3 | 17.2 | 22.1 | 23.4 | 28.3 | 35.8 | 27.5 | 31.1 | 31.5 | 34.4 | 37.2 |
| Ours | **37.8** | **40.3** | **41.7** | **47.3** | 49.4 | **41.6** | **43.0** | **43.4** | **47.4** | **49.1** | **33.3** | **38.0** | **39.8** | **41.5** | 44.8 |

present the results of different intervals ($\tau$, $t$) in Table. 2-Right (bottom). This interval controls the difficulty of negative proposals: if the bounds are set too high (e.g. (0.3, 0.4)), it becomes too difficult for the network to distinguish negatives from positives, the performance drops clearly (e.g. 58.6). On the other hand, if the bounds are set too low (e.g. (0, 0.1)), there may not be enough hard negatives for the network to learn, the performance is also not optimal (e.g. 68.0). Our default setting of (0.2, 03) offers a moderate way to generate hard negatives.

In general, NP-RepMet is insensitive to neither IoU or $\beta$. The performance drop for varying IoU or $\beta$ is not much within certain ranges; and it is always better than not using it (e.g. $\beta = 0$). This suggests that it is helpful to mine negative information within the images of the same object class.

## 4.4 Results on PASCAL VOC 2007

Following the training and testing details in Sec. 4.2, we conduct experiments on PASCAL VOC 2007 and compare our method with other SOTA [10–12, 1][2]. Results are in Table 3 on the three novel/base class splits. One can clearly see that ours outperforms the SOTA in almost every entry and the improvements are also consistent for different base/novel class splits and number of shots. The improved margin is particularly large when the labels are extremely scarce (1-3 shot). For instance on the second split, our NP-RepMet produces a very high mAP 41.6 in 1-shot which is 19.8% higher than the previous best result (21.8%). When the support set size for novel classes increases, the number of positive proposals in the set becomes more diverse, hence the effect of mining negative information reduces. Similar observation was also found on the ImageNet-LOC experiment (Table 1). This suggests our idea of restoring negative information as a vital technique for the real few-shot scenario. We also report the 3- and 10-shot results on the base classes of the first split. Results in Table 4 show that our NP-RepMet delivers a good detection performance on base classes as well.

Notice: 1) Our NP-RepMet produces lower mAP than [11] in 10-shot on the first and third base/novel class split. Yet, it appears that [11] uses different support sets for novel classes from [10, 12] and us. Different support sets would affect the performance: we observed better results of NP-RepMet when we varied the support set. 2) We let our NP-RepMet detect the base and novel classes simultaneously on PASCAL VOC. Yet, we did not find clear evidence among [10–12] that they all did the same thing ([10, 11] seem to be the same with us). We want to point out, if we let NP-RepMet detect novel

Table 4: 3/10-shot performance on PASCAL VOC 2007 novel and base classes of the first split.

| Shot | Method | Novel classes | | | | | | Base classes | | | | | | | | | | | | | | |
|---|---|---|---|---|---|---|---|---|---|---|---|---|---|---|---|---|---|---|---|---|---|---|---|
| | | bird | bus | cow | mbik | sofa | mean | aero | bicy | boat | bottl | car | cat | chair | tabl | dog | horse | pers | plant | sheep | train | tv | mean |
| 3 | [10] | 26.1 | 19.1 | 40.7 | 20.4 | 27.1 | 26.7 | 73.6 | 73.1 | 56.7 | 41.6 | **76.1** | 78.7 | 42.6 | **66.8** | 72.0 | **77.7** | 68.5 | 42.0 | 57.1 | **74.7** | **70.7** | 64.8 |
| | [11] | **30.1** | 44.6 | **50.8** | 38.8 | 10.7 | 35.0 | 67.6 | 70.5 | 59.8 | 50.0 | 75.7 | **81.4** | 44.9 | 57.7 | **76.3** | 74.9 | **76.9** | 34.7 | 58.7 | **74.7** | 67.8 | 64.8 |
| | Ours | 12.9 | **60.5** | 39.9 | **43.1** | **52.2** | **41.7** | **79.8** | **82.5** | **66.9** | **73.8** | 71.6 | 57.6 | **52.9** | 64.1 | 49.6 | 70.7 | 71.8 | **58.7** | **74.2** | 55.0 | 69.5 | **66.6** |
| 10 | [10] | 30.0 | 62.7 | 43.2 | **60.6** | 39.6 | 47.2 | - | - | - | - | - | - | - | - | - | - | - | - | - | - | - | 69.7 |
| | [11] | **52.5** | 55.9 | **52.7** | 54.6 | 41.6 | **51.5** | 68.1 | 73.9 | 59.8 | 54.2 | **80.1** | 82.9 | 48.8 | 62.8 | **80.1** | **81.4** | **77.2** | 37.2 | 65.7 | **75.8** | 70.6 | 67.9 |
| | Ours | 18.2 | **64.5** | 51.3 | 57.3 | **55.7** | 49.4 | **71.9** | **79.1** | **64.9** | **70.8** | 73.6 | 49.5 | **53.5** | **67.3** | 62.7 | 78.7 | 74.8 | **58.3** | **76.2** | 72.5 | 67.9 | **68.3** |

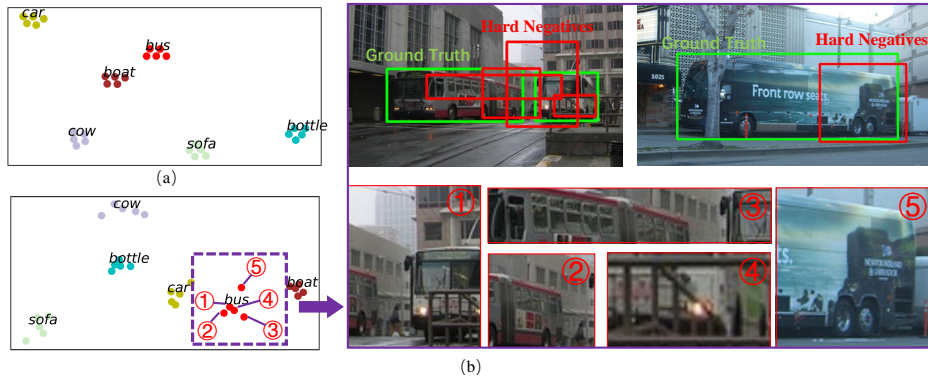

(a)

(b)

Figure 4: Distribution of positive and negative class representatives in NP-RepMet on PASCAL VOC 2007. (a) T-SNE visualization of positive class representatives. Different colors indicate different classes; (b) T-SNE visualization of negative class representatives and the selected hard negative proposals from the support images (3-shot) of a specific class *bus*.

classes only, without the interference of base classes, its results can be further improved; for instance, in Novel Set 1, we achieve mAP *39.3*, *43.7*, *45.9*, *51.7*, and *56.2* in 1, 2, 3, 5 and 10-shot, respectively.

Table 4 also shows the individual class AP (average precision) for both base and novel classes in the first base/novel split.

Last, similar to [10, 11], we use t-SNE [40] visualization to demonstrate the distribution of positive and negative class representatives in NP-RepMet. Figure 4(a) illustrates the positive representatives of a subset of classes on PASCAL VOC. The representatives of base classes are learnt in the model while the representatives of novel classes are obtained via the embedding vectors of positive proposals from support images. The results show that the representative vectors well depict each class by clustering together for the same class and repulsing those from the other classes. Figure 4(b) draws the embedded negative representatives of the subset of classes while focuses on the selected hard negatives from a specific class *bus*. Negative representatives of different classes are clearly distinguished from each other. Given a query, its hard negative proposals (e.g. partial object) for certain class (e.g. bus) can be easily filtered out by comparing to those negative representatives from the support set.

## 5   Conclusion

In the regime of few-shot learning, few-shot object detection has not been largely explored. Representative works tend to focus on the foreground (positive) area of images, such that matured few-shot classification techniques can be easily applied or adapted. In this paper, we propose to restore the negative information in the few-shot object detection: we show that hard negatives are essential for the metric learning in few-shot object detection. We build our work on top of a state-of-the-art pipeline, RepMet, where we introduce several new modules such as negative and positive representatives, NP-embedding, triplet losses based on NP-embedding, etc. A new inference scheme is also introduced given the learnt negative representatives in the pipeline. We conduct extensive experiments on two standard benchmarks, ImageNet-LOC and PASCAL VOC 2007. Results show that our method significantly improves the SOTA by simply restoring negative information into it.

## Broader Impact

Object detection is one of the most fundamental tasks in computer vision field, it serves as a key component for downstream algorithms and applications, including instance segmentation, human pose estimation and tracking. The majority of deep learning methods are designed to solve fully-supervised problems which drives the demand and progress of few-shot learning in object detection.

The proposed method is designed to solve few-shot detection problem, researchers focus on high-level recognition tasks may benefit from our work. There may be unpredictable failures, similar as most other detectors. Please do not use it for scenarios where failures will lead to serious consequences. The method is data driven, and thus the performance may be affected by the biases in the data. So please also be careful about the data collection process when using it.

## Acknowledgments and Disclosure of Funding

This work was partially supported by the National Natural Science Foundation of China (NSFC) under Grant No. 61828602; the National Key R&D program (2018YFE0200200); the Grant from the Institute for Guo Qiang of Tsinghua University and Beijing Academy of Artificial Intelligence (BAAI); the Science and Technology Major Project of Guangzhou (202007030006); the open project of Zhejiang Laboratory.

## Footnotes

\*The work is done when Yukuan Yang was an intern at Microsoft Research Asia. Miaojing Shi and Guoqi Li are the corresponding authors.

[2]We reproduce RepMet [1] on PASCAL VOC 2007 ourselves.

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
