[Reviews · NeurIPS 2020]

Review 1

Summary and Contributions: This paper proposes an algorithm for few-shot object detection based on the insight that hard negatives (namely badly localized boxes in the support set) are important. The proposed approach is shown to outperform prior art.

Strengths: - The insight that badly localized boxes are much needed negatives is a great one, and very important. - The results are impressive and clearly beat current statee-of-the-art. - The approach is clearly novel. - Few-shot object detection is a problem of great practical significance, so these results are important.

Weaknesses: (1) If the insight is that hard negatives are important, then a very simple baseline presents itself: why not take a simple object detector trained on the base classes, and simply finetune the detector head and bbox regressor head in the usual way on the novel classes? This would automatically use the badly localized examples. I am surprised that the authors did not include this baseline. (2)The numbers reported in Table 4 all use different network backbones. YOLO-FR uses DarkNet-19, Meta-Det uses VGG16, Meta-RCNN uses ResNet-101 (but without FPN or DCN). Meta-Det for e.g. pretrains the backbone on ImageNet. RepMet pretrains it on Coco. It is unclear what this paper pretrains it on. These differences are absolutely crucial and can easily explain any of the observed differences in performance. Ideally, the comparison would be apples to apples using identical backbones and pretraining regimes. But I accept that this may be too difficult. But at the very least, the paper should be explicit about exactly what the backbones and settings of each compared technique is. I cannot imagine accepting the paper without this, since without controlling for backbones and pretraining regimes, the conclusions are basically meaningless. (3) While this is not necessary for acceptance, I highly recommend the authors show results on Coco and/or LVIS, since they are much harder than Imagenet-LOC or Pascal VOC.

Correctness: See above.

Clarity: Yes.

Relation to Prior Work: Yes.

Reproducibility: No

Additional Feedback: I want to see points (1) and (2) addressed for me to recommend acceptance. [EDIT]: Upgraded review based on rebuttal.


Review 2

Summary and Contributions: The goal of this paper is to restore the information in negative proposals for few-shot object detection. The proposed method is built on the pipeline of RepMet by modifying and adding several modules to learn the embedding space and the representatives from positive and negative proposals using triplet losses. The main focus of the paper is on describing how the modifications can improve RepMet. To reduce the number of negative proposals, the paper proposes to use spectral clustering for choosing negative and positive representative proposals from the support images. The results show that the proposed method significantly outperforms RepMet on ImageNet-LOC and PASCAL VOC benchmarks for few-shot object detection.

Strengths: The writing of the paper is clear. The idea of including another branch in RepMet to learn negative embeddings from negative proposals seems to work well for object detection, since in object detection the issue of foreground-background imbalance is more essential and critical than in classification.

Weaknesses: The main idea introduced by the proposed method is to add negative representatives into RepMet. Although this modification seems to be effective, the novelty and contribution might be limited, because the problem that is addresed by this paper is specific for RepMet, due to the fact that RepMet does not take the negative proposals into consideration for learning the embeddings. Other few-shot methods like [16], MetaDet, and Meta R-CNN do not rely on representatives and hence do not suffer from this problem. It is reasonable and easier to understand why modeling the postive representatives is useful for few-shot object detection. However, the idea of learning 'negative-representatives' is a little strange since the diversity of background is much larger, and it might be less intuitive that a set of meaningful 'negative-representatives' can be learned from the hard negatives extracted from a few sample images and then can be used to distinguish the negative proposals in a query image that exhibits a very different background. It would be helpful if some examples of Cluster-Min can be visualized to provide some insights on what negative-representatives are learned.

Correctness: Because the backbone is pre-trained (using ImageNet or COCO), it is necessary to describe in detail whether the pre-trained data include samples of the novel classes. The RPN takes features from the backbone, and hence if the pre-trained backbone has "seen" some samples of novel classes, the underlying features could be more sensitive to the target foreground objects even if the RPN is trained from scratch. Although the experimental settings in this paper follow previous works like RepMet, MetaDet, and [16], the issue of data contamination through the pre-trained backbone should be discussed when evaluating few-shot learning methods. Triplet/ranking/margin losses are common in the literature of object detection and metric learning. It is practical to use the triplet loss for learning NP-embedding.

Clarity: Yes, the paper is well written.

Relation to Prior Work: The following NeurIPS paper on one-shot object detection is related and can be considered for comparison: Hsieh, T.I., Lo, Y.C., Chen, H.T., Liu, T.L.: One-shot object detection with co-attention and co-excitation. In: NeurIPS (2019) Although the experimental settings might be different, it seems that their method achieves better results on VOC dataset than the proposed method for novel classes. For the design of DML module, the following CVPR paper can also be referred to: Wang, X., Hua, Y., Kodirov, E., Hu, G., Garnier, R., Robertson, N.M.: Ranked list loss for deep metric learning. In: CVPR (2019)

Reproducibility: Yes

Additional Feedback: Typos: L111: denotes --> denote L258: v.s. --> vs. L291-L293: add space between "e.g." and the number


Review 3

Summary and Contributions: The work extends a distance-based few-shot detector by adding a new type of hard negative samples, which are regions poorly intersecting the same ground truth region as the positive sample (in addition to hard negatives of other classes). When applied to the RepMet detector, this leads to substantial performance boost.

Strengths: The work proposes a new principle in hard negatives selection for the triplet loss based learning of object detector with distance-based region classifier (e.g., RepMet). The proposed regions with 0.2<IoU<0.3 are declared as special type of hard negatives and a special attention is given to them by learning negative prototypes (representatives) for these regions. They are used in inference as well as the training. The approach is shown to achieve SOTA performance on two benchmarks, boosting the original RepMet (on Imagenet-LOC) and improving upon other detectors (on PASCAL VOC)

Weaknesses: Two obvious questions, that come to mind, were not answered in the paper. 1. The proposed method is not restricted to few-shot regime, it can be used to train a standard detector (which is one of RepMet modes). How would it rate in this scenario? 2. Why weren't the standard RepMet applied to the second benchmark (Pascal VOC), along with its extended version (the NP-RepMet)? I would expect to see these omissions filled (or explained, in case I am missing something).

Correctness: The claims, method and methodology, described in the paper seem correct.

Clarity: The paper is written with clarity and good language.

Relation to Prior Work: The relation to prior work is properly discussed.

Reproducibility: Yes

Additional Feedback: I encourage the authors to test the NP-RepMet in the standard training regime, as the scope of proposed technique may be wider than just the few-shot scenario. Also I would like to see RepMet's performance on Pascal VOC (which is obviously within technical capacity of the authors). My comments were addressed in the authors response, I therefore maintain my score.


Review 4

Summary and Contributions: In this paper, the authors have introduced a new idea of the restoration of negative information in few-shot detection.They proposed a new negative- and positive-representative based metric learning framework with negative information incorporated in different modules for better feature steering in the embedding space.

Strengths: - The authors have clearly presented the motivation of their method in Figure 1. - They have conducted extensive experiments to demonstrate the effectiveness of their approach.

Weaknesses: - For experimental results in Table 4, the authors did not report the results of [16] for 3-shot. From the results, can we conclude that the proposed method does not have better performance than [16]? The authors fail to provide a detailed analysis on this. - The authors take IoU>0.7 as positives and 0.2<IoU<0.3 as negatives. How does this parameter determined? This is not included in the current version. - The authors fail to compare against state-of-the-art hard negative mining methods in the literature, although they have not been directly applied to few-shot object detection.

Correctness: Yes

Clarity: Yes

Relation to Prior Work: Yes

Reproducibility: Yes

Additional Feedback:

[Author Response · NeurIPS 2020]

Figure 1: Distribution of embedded negative representatives using t-SNE; The selected hard negative proposals from support images (3-shot, i.e. 3 *buses*) of *bus*. [Zoom In]

| Shot | 1 | 2 | 3 | 5 | 10 |
|---|---|---|---|---|---|
| Novel Set 1 | 26.1 | 32.9 | 34.4 | 38.6 | 41.3 |
| Novel Set 2 | 17.2 | 22.1 | 23.4 | 28.3 | 35.8 |
| Novel Set 3 | 27.5 | 31.1 | 31.5 | 34.4 | 37.2 |

Table 1: Performance of RepMet on PASCAL VOC novel sets.

We thank all reviewers for their constructive and valuable comments.

**(R1) Baseline: training a simple object detector on base classes and fine-tuneing its head on novel classes.**

A1: Indeed, the proposed baseline has already been implemented in RepMet[17] (see its Page 6 and Table 3 for "baseline-FT"). This baseline on ImageNet-LOC performs much inferior than RepMet/NP-RepMet. We will add this.

**(R1) The backbone and pre-training regime of the network should be controlled for comparison.**

A2: We agree ! We mentioned this in Sec. 4.2: we use ResNet-101 as the backbone (L213); its weights initialization follows [17] for ImageNet-LOC and [16] for PASCAL VOC (L215); weights of other modules (e.g. FPN, RPN) are randomly initialized (L216). That is, we pre-train the backbone on COCO to fairly compare with RepMet on ImageNet-LOC; pre-train it on ImageNet to align with [16, 36,38] on PASCAL VOC. We are also very careful to follow train/test details (class splits (L211-212), support sets (L224-226), inference scheme (L311-319)) in [17] and [16].

**(R2) [16,36,38] do not rely on representatives and hence do not suffer from the problem addressed in this work.**

A3: The key observation of this work is that hard negative information within support images is not carefully exploited in previous works [16,17,36,38]. They simply extracts positive proposals w.r.t. ground truth from support images while negatives proposals containing e.g. partial objects or ambiguous surroundings are not considered (L37). These negatives however are important false positives in object detection (L40-43). Although we build our work on RepMet, our idea of restoring negative information should be essential and beneficial to many few-shot detection works. Judging from the results, NP-RepMet drastically outperforms other few-shot methods [16, 36, 38] on PASCAL VOC (e.g. up to 18%).

**(R2) It is less intuitive that negative representatives can be used in a query image with different background.**

A4: First, we emphasize that hard negative proposals mined in this work are mainly focused on proposals containing partial, occluded object or entire object with massive surroundings. To realize this, we propose to specifically choose them via IoU thresh (0.2, 0.3) and Cluster-Min. Fig. 1 illustrates the selected hard negatives from the support set of *bus*. We also use t-SNE to draw the embedded negative representatives from the support sets of multiple classes: different classes are clearly distinguished from each other; given a query, its hard negative proposals (e.g. partial object) for certain class can be easily filtered out by comparing to those negative representatives from the support set. We provide ablation study in Sec 4.3 and Table 2 to show that these negative representatives contribute substantially to NP-RepMet.

**(R2) Discussion on whether the pre-trained data include samples of the novel classes.**

A5: Our pre-training follows previous works (see R1-A2). Class overlap between the pre-trained data and novel data may exist, noting that ImageNet contains 1000 classes. Yet, we argue 1) the class distribution on pretrained data is very different to that on novel data; 2) detection-related modules in NP-RepMet are randomly initialized; 3) as long as base class data are enough, we do not think using a pretrained model would cast a strong impact on the final performance.

**(R2) More refs.** A6: Thanks! We will carefully discuss them! Particularly, the NeurIPS paper indeed has very different train/test settings (class splits, support sets, queries, etc.) to ours and [16,38,36], so the results are not comparable.

**(R3) The proposed method is not restricted to few-shot regime, it can be used to train a standard detector**

A7: Yes! This can be validated from results in Table 1, 4 in the paper and Table 1 in the supp, where we train the detector on the train set of base classes and evaluate it on the test set of base classes to show it maintains a good accuracy; some comparison to standard detectors can also be found in Table 3 in [38]; notice our NP-RepMet is inferred for all the classes together on PASCAL VOC (L315-319), like a standard detector. On the other hand, if R3 meant fine-tuning NP-RepMet with novel classes instead of meta-testing as in the paper, yes, we can further fine-tune it and our results can be improved to 71.3, 78.6, and 81.7 for 1, 5, and 10-shot on ImageNet-LOC unseen (novel) classes.

**(R3) Performance of RepMet on PASCAL VOC.**

A8: RepMet did not report results on PASCAL VOC novel sets. We can reproduce it under the same setting with ours, results are in Table 1: RepMet performs clearly inferior to our NP-RepMet (Table 3 in the paper).

**(R4) The authors did not report the results of [16] for 3-shot base classes.**

A9: It is because [16] did not report it in their paper. One might find it being reported in [38] as 64.8 which is lower than our 66.6; but we also found that [38] reports the 10-shot result of [16] as 63.6 instead of 69.7. Overall, the base class results among NP-RepMet and [16,38] are competitive to each other and all maintain a good accuracy.

**(R4) IoU hyper-parameter.** A10: Taking IoU>0.7 for positive samples is a common practice for many detection works, e.g. Faster R-CNN and RepMet; we simply follow this. We experimented with IoU>0.5, the result is 68.3 (v.s. 68.5 for IoU>0.7, very close) on the 1-shot case of ImageNet-LOC. As for negative sample selection, we perform ablation study for different IoU in Table 2 (right bottom), this hyper-parameter is quite robust within certain range.

**(R4) No comparison to other hard negative mining methods, though they have not applied to few-shot detection.**

A11: One representative hard negative mining strategy OHEM [30] is indeed used in RepMet (L101) where class representatives from different classes are considered as negatives to each other. We actually adopt this in our loss function (Eq 1 and 2). We bootstrap the classifier with negative information both within and across images and classes. More importantly, we show that the idea of mining hard negatives within the same image of positives is essential to few-shot detection. We will add more discussion about hard negative mining in literature in the revised version.

[Meta-Review · NeurIPS 2020]

The reviewers have supported the acceptance of this paper but noted the novelty is somewhat limited. It would be great to highlight how the proposed technique can be used outside of the RepMet framework.